# Mobility Coupled with Motivation Promotes Survival: The Evolution of Cognition as an Adaptive Strategy

**DOI:** 10.3390/biology12010080

**Published:** 2023-01-03

**Authors:** George B. Stefano, Richard M. Kream, Tobias Esch

**Affiliations:** 1Institute for Integrative Health Care and Health Promotion, School of Medicine, Witten/Herdecke University, 58455 Witten, Germany; 2Department of Psychiatry, First Faculty of Medicine, Charles University and General University Hospital in Prague, 120 00 Prague, Czech Republic

**Keywords:** dopamine, catecholamines, morphine, evolution, mobility, cognition, reward, motivation, behavior, stress, μ3 receptor

## Abstract

**Simple Summary:**

Here we present the hypothesis of an evolutionary and functional relationship between the occurrence and use of the catecholamine dopamine (DA) as a neurotransmitter (messenger)—particularly in invertebrates—and the catecholamines epinephrine (EP) and norepinephrine (NE), messengers that are found only in vertebrates. Interestingly, both are also involved in pathways leading to the production of endogenous morphine, another messenger substance. We assume that the use of EP/NE as messengers represents an evolutionary advantage and adaptation process, whereby this “metabolite” (its biochemical intermediates) is only used “in retrospect” as a neurotransmitter (evolutionary “retrofitting”); on the way to greater mobility, with a need to expand data storage (memory, cognition) within the scope of this expanded radius, additional messengers were needed. Moreover, challenges and “stress” coming with increased mobility (e.g., entering unfamiliar environments) had to be successfully met to ensure survival. The same applies to the synthesis of morphine, which is formed from tyramine and tyrosine via DA (mediated by enzymes that also interact with EP/NE) so that morphine can be chemically classified as an “end product” of a DA-opiate cascade. Morphine’s functional importance is the downregulation and termination of a motivational sequence from wanting (appetite) to avoiding (avoidance) to relaxation/quiescence (assertion).

**Abstract:**

Morphine plays a critical regulatory role in both simple and complex plant species. Dopamine is a critical chemical intermediate in the morphine biosynthetic pathway and may have served as a primordial agonist in developing catecholamine signaling pathways. While dopamine remains the preeminent catecholamine in invertebrate neural systems, epinephrine is the major product of catecholamine synthetic pathways in vertebrate species. Given that the enzymatic steps leading to the generation of morphine are similar to those constraining the evolutionary adaptation of the biosynthesis of catecholamines, we hypothesize that the emergence of these more advanced signaling pathways was based on conservation and selective “retrofitting” of pre-existing enzyme activities. This is consistent with observations that support the recruitment of enzymatically synthesized tetrahydrobiopterin (BH4), which is a cofactor for tyrosine hydroxylase, the enzyme responsible for dopamine production. BH4 is also an electron donor involved in the production of nitric oxide (NO). The links that coordinate BH4-mediated NO and catecholaminergic-mediated processes provide these systems with the capacity to regulate numerous downstream signaling pathways. We hypothesize that the evolution of catecholamine signaling pathways in animal species depends on the acquisition of a mobile lifestyle and motivationally driven feeding, sexual, and self-protective responses.

## 1. Introduction

Responses to many complex evolutionary constraints leading to the development of advanced biological systems involve the preservation of current pathways as well as their use as templates to promote the growth and development of multifunctional chemical motifs. The biosynthesis of L-tyrosine (L-TYR), which is a biosynthetic precursor of the biochemical neurotransmitter, dopamine, is an important example of combinatorial chemistry as it relates to evolution. The synthetic process leading to L-TYR includes the incorporation of a phenolic functional group as a chemical side chain. This modification provides several unique chemical advantages to this amino acid as it enhances its capacity to function in both animal and plant cells. The evolution of enzymes that modify free L-TYR has reinforced our understanding of the preeminence of this amino acid as a precursor species in catecholamine biosynthetic pathways. L-TYR is also a precursor in the morphine biosynthetic pathway described in the opium poppy (*Papaver somniferum*) [1].

Phytoalexin, a derivative of morphine, serves as an antimicrobial agent in the opium poppy; dopamine (DA) may be a biosynthetic intermediate in its synthesis [2]. Similarly, catecholamine and other “morphinergic” signaling systems have evolved as critical regulators of pathways in animal species [2] including those involved in energy metabolism, homeostasis, and mitochondrial respiration. These pathways share a common set of biosynthetic and metabolic enzymes, suggesting that this process may result from the “retrofitting” of several ancient enzyme species [1].

The second example of combinatorial chemistry associated with the evolutionary process focuses on L-arginine (L-ARG) [3,4,5]. Oxidation of the L-ARG guanine group followed by its cleavage results in the release of L-citrulline and nitric oxide (NO); this conserved pathway contributes to a critically important cellular signaling mechanism in both plants and animals [6,7,8]. Formation and release of constitutive NO are mediated by Ca^2+^-dependent NO synthases (NOSs) [9].

The critical metabolic and signaling roles played by both L-TYR and L-ARG are highlighted by the interactive regulatory activities of DA, morphine, and NO in diverse biological systems (Figure 1). These ancient regulatory relationships have been conserved in order to maintain the dynamic equilibrium between nitrogen and oxygen. Several evolutionary, developmental, and regulatory pathways that include DA, “morphinergic,” and NO-mediated signaling are described further below.

## 2. Morphine and Dopamine as Signaling Molecules

### 2.1. Evolutionary Significance of Catecholaminergic Signaling

The maturation and release of morphine and benzylisoquinoline (BIQ) alkaloids take place in *P. somniferum* cells. DA is used exclusively in plant species as a substrate for Pictet-Spengler condensation with tyramine aldehyde to generate norcoclaurine, which is the first committed chemical intermediate in morphine biosynthesis [22,23,24,25,26,27,28,29,30,31,32,33,34,35,36].

All invertebrate phyla retain the capacity to synthesize morphine [10,11,37]. However, catecholamines (including DA) have emerged as signaling molecules in numerous invertebrate organ systems [12,13,38,39]. We were unable to detect epinephrine (EP) in invertebrate tissues; this result suggests that sequential methylation and hydroxylation of DA leading to the formation of EP developed later during evolution. By contrast, the catecholamine pathway has undergone extensive development in vertebrate species and terminates in the biosynthesis of EP [14].

In our previous publications, we highlighted the emergence of DA as a prototype signaling molecule and discussed its critical role in dopaminergic/catecholaminergic signaling pathways [2]. As noted above, while both plants and animals can synthesize morphine, the catecholamine pathway and regulated expression of EP exist only in vertebrate species. These findings suggest that catecholamine and related signaling systems emerged from the morphine-based pathway via the adaptation of enzymes involved in L-TYR, levodopa (L-DOPA), DA, and tyramine modification [1,2,11,15]. Morphine promotes the downregulation of tissue excitability via interactions with specific receptors (see below), thereby protecting tissues from hyperexcitability [9,16,40,41]. With the development of motor activities leading to feeding, sexuality, and protective responses characteristic of invertebrates, a new DA-dependent signaling system emerged.

Thus, DA signaling is associated with newly developed mobility and motivational systems characterized in invertebrate species [2]. We hypothesize that the successful adaptation of dopaminergic regulatory functions in invertebrates led to its continued use and development in vertebrate species (Figure 1; see below). This process included full incorporation of NE and EP signaling pathways underlying essential motor processes that promote greater mobility in support of food acquisition while simultaneously contributing to motivational signaling, e.g., hunger [2]. Thus, the initial system that promoted motor activation and its associated behaviors expanded to include complex motor activities as well as highly developed motivational processes (i.e., reward, pleasure, and pain) [2]. Thus, the well-established pathways that link motor activities with emotional processes via a set of shared chemical messengers act synergistically (Figure 2). We hypothesize that the emergence of catecholaminergic signaling systems that mediate complex emotional and cognitive processes developed as a means to focus the activated motor processes as a superior survival strategy [2,42,43].

We also hypothesize that the emergence of catecholaminergic signaling was facilitated by the “retrofit” of enzymes that contributed to morphine biosynthesis. This process accommodated biochemical maturation and modification of catecholamines related to DA [1,2,13,17]. This hypothesis also incorporates our current understanding of NO-mediated cellular signaling associated with morphinergic/catecholaminergic regulatory processes; this aspect remains critical because NO can be detected in all vertebrate and invertebrate phyla [9,45,46,47]. Moreover, recent critical studies have identified mitochondrial NOS (mtNOS)-derived NO as capable of regulating oxygen and energy metabolism without generating oxidative stress [3,48]. Taken together, our description of a morphine synthesis pathway in animal cells with characteristics that were similar to those identified in *P. somniferum* as well as morphine’s capacity to stimulate the release of NO provides an outline of the evolutionary constraints acting on ancient regulatory circuitries [1]. These results also suggest that this system was developed and carried over from a common ancestor of plants and animals that presents itself as a key mediator of energy and developmental processes [49].

### 2.2. Role of the μ3 Opioid Receptor

As further support for our hypothesis, our previous studies highlighted the role of a unique opiate receptor expressed by the MOR (μ opioid receptor) gene [18,50,51]. This receptor (designated μ3) is a Class A rhodopsin-like member of the G-protein coupled receptor superfamily that mediates the effects of morphine and related alkaloids via activation of constitutive NO production and release [9,16,19,41,45,50]. The μ3 receptor binding profile includes morphine but not opioid peptides, which is why it can be classified as morphine-specific, i.e., linked to morphinergic (opiate alkaloid) signaling. The μ3 opiate receptor has been detected specifically in human stem cells [19]. MOR and constitutive NO release may contribute to a mechanism used to “manage” evolving catecholaminergic signaling, specifically to control processes associated with catecholamine activation (i.e., feeding, movement, sexual activity). Thus, we speculate, that these originally “calming” homeostatic processes later emerged as capable of selective activation and subsequent downregulation once the goal has been accomplished.

### 2.3. Dopamine and Morphine Signaling in Plants

The endogenous expression of DA [52] and/or its related amino acid precursor L-DOPA [53] has been demonstrated in plants to serve as an existential phytoalexins designed to protect the plant against other invasive plant species. For example, L-DOPA is a major plant phytotoxin capable of inhibiting nearby plant growth following release into the soil [53]. The functional role of DA released from plants appears to be more complex and mediates protection or tolerance against diverse abiotic stressors associated with drought, high salt, and nutrient deprivation as well as positively affecting immune responses against plant diseases [52]. Furthermore, DA appears to represent an endogenous regulator of plant development and stress-related gene expression. Interestingly, in opium poppy, stress signals have been shown to promote formation of a bismorphine dimer from two morphine subunits, followed by accumulation in the cell wall via binding to pectin [54]. The authors hypothesize that rapid formation of a toxic metabolite of morphine in the form of bismorphine enables the opium poppy to rapidly mount a defensive mechanism against diverse microbial assaults potentially leading to subsequent destruction of the plant. Taken together, it appears both “linked” signaling molecules serve useful important functions in plants as in animals [17]. The association of plant and animal signaling of DA, L-DOPA, and morphine with stress substantiates their ability to evolve even a stronger influence in the evolution of mobility as well as being involved in motivational signaling.

### 2.4. Reward Systems Involvement and Behavioral Responses in Animals

The biological mechanisms responsible for complex behaviors in animals are motivated by pleasurable events known as “rewards”. These processes are mediated by the limbic system in the brain [20,44,55,56,57,58,59,60,61,62] and depend largely on dopaminergic signaling [56]. Recent studies have revealed a link between endogenously expressed morphine and reward-directed behavioral processes [16,37,63,64]. Other neurotransmitters, including glutamate, gamma-aminobutyric acid (GABA), stress hormones, and serotonin also play critical roles in this process [65,66,67]. Naturally rewarding experiences may also activate the brain’s motivation and reward circuitries [55,67,68,69]. Psychomotor stimulants (e.g., amphetamine, cocaine) and opiates (e.g., heroin, morphine) may also activate the reward circuits. The capacity of these and other addictive drugs to promote strong activation of central nervous system (CNS) reward systems and alter their normal functions are among the crucial features of addiction/substance abuse [40,55,70,71,72,73]. Alterations in reward circuitry associated with drug abuse and pleasure-seeking behavior may promote tolerance, dependence, craving, relapse, and increased vulnerability [56,62,74].

Results from numerous previous studies revealed that drugs of abuse can promote DA release from the ventral tegmental area (VTA) into the nucleus accumbens. This will alter the glutamate responsiveness of the prefrontal cortex [56,61,62]. Changes in sensitivity to glutamate may augment DA release from the VTA and increase the responses to DA in the nucleus accumbens, thereby promoting delta FosB and CREB gene activity [56,74]. During periods of prolonged abstinence, changes in gene activity and signaling can also be observed [61,62,74,75,76]. These actions may trigger relapse by increasing drug sensitivity by eliciting powerful responses and cues [56,61,77]. Counterintuitively, abstinence from cocaine or morphine after a period of repeated use may result in a decrease in DA levels in the mesolimbic DA system/VTA [78,79]. Compromised dopaminergic signaling may be associated with food cravings seen during opiate withdrawal in human subjects [55].

Addictive drugs can also stimulate the dopaminergic reward system. For example, heroin enhances DA levels by promoting its increased release, while cocaine inhibits DA reuptake; these actions have been related to their impact on appetitive motivation [55,56,80]. Opiates may also generate “reward messages” via direct actions at the nucleus accumbens [56,81]. Interestingly, DNA, RNA, cocaine, nicotine, and dopamine are all considered to be alkaloids and thus share an important evolutionary commonality [82]. Additionally, several published studies have reported links between dopaminergic and “morphinergic” signaling pathways [1,11,15]. As noted above, DA is involved in the production of endogenous morphine [1,11,15]. Therefore, altered DA metabolism may have an immediate impact on morphine biosynthesis and thus weaken the actions of the endogenous compound on the “DA” reward system [37,67,83].

The effects of morphinergic and dopaminergic regulatory processes can lead to complex behavioral endpoints. From this perspective, hyperactivity, aggression, and rage—as examples—may be followed by a period of limited activity [2,70]. This biphasic response (i.e., hyperactivity followed by relaxation) may have important evolutionary value (see below). The state of hyperawareness that is regulated by dopaminergic processes may be associated with a significant survival advantage [84,85,86] as it would permit an organism to be prepared for unexpected and potentially life-threatening events. Once this type of threat is deemed or perceived as nonexistent, relaxation mediated by these processes may result [17]. This combination of events may be the result of different manifestations of integrated morphinergic and dopaminergic signaling processes.

### 2.5. Mobility and Cognition

Seen from an evolutionary perspective, the development of motor behavior and mobility in higher organisms was associated with a great advantage: New habitats could be entered and developed and with them additional sources of food and space for the growth and spread of one’s own species. However, a successful adaptation to these new habitats and their novel opportunities was required—in addition to improved psychomotor skills for actually getting there (starting, e.g., with the decision to move, i.e., “fantasy”, spontaneity; followed by the actual implementation of the new behavior, i.e., factual movement into these spaces). These processes included a state of heightened alertness and vigilance to perceive and recognize—and fight—new and unfamiliar challenges that likely occurred, i.e., stress [58,87] (Figure 3).

Moreover, in case of an individual (or species) actually moving into new and yet “uncharted” territories or habitats, their (collective) nervous systems had to be able to remember the path (the “map”) that brought them here; i.e., they had to be able to locate and pinpoint their current position in order to be able to return at any time. In this respect, indeed, intact and effective stress physiology (associated with well-tuned relaxation mechanisms, see above) and—besides the psychological as well as physical mobility needed to get there—improved memory skills were required. Some evolutionary biologists believe that these very “requirements” were essential drivers for the evolutionary growth in the size of the brains of higher animals [88,89,90,91,92,93].

Hence, mobile creatures had to evolve an alert state (stress, survival), which later added the ability to determine the significance of this alert state and develop solutions (memory, cognition, intelligence [87,88]). For this, in the beginning, a “desire” for change was required (→ fantasy, appetitive motivation [94]), followed by the actual psychomotor impulse to move (spontaneity) and implement the change (→ exert mobility, motor behavior [94]), which also required an overcoming of neophobia and a general ability to meet constant challenges occurring on the path (→ stress response: higher alertness, threat avoidance, fight-or-flight [87,94]). Finally, the ability to rebalance and recover from this stress cascade was critical (→ relaxation response), also learning from its “adventures” for future encounters (→ cognition), and yet terminating this neurobiological sequence by returning to normal and being “happy”, in the present moment [42,87,88,94,95]. Interestingly, for the said sequence to be performed, catecholamines (e.g., dopamine: psychomotor activities; adrenaline/noradrenaline: stress response), as well as endogenous morphine (down-regulation, termination), seem neurobiologically best suited (Figure 1, Figure 2 and Figure 3).

## 3. Conclusions

In summary, while both plants and animals can generate endogenous morphine, only the catecholamine signaling pathway is used in vertebrate species. Both catecholamine and endogenous morphine biosynthesis rely on tyrosine, L-DOPA, DA, and tyramine; these findings strongly suggest that catecholamine biosynthetic pathways emerged from morphine-directed precursors [1,11,15].

The advantages conferred by this new signaling pathway are not immediately clear. Based on findings published in numerous previous reports, we hypothesize that morphine promotes overall downregulation and protects against tissue hyperexcitability via its interactions with specific cellular receptors in both animals and plants [9,16,24,40,41]. Catecholamine signaling pathways developed to support complex activities in both vertebrates and invertebrates, including those associated with feeding, sexuality, and protective responses; DA may serve as a critical signaling molecule in these animals.

We also hypothesize that dopaminergic regulatory functions that successfully promoted mobility in invertebrates were carried over and developed further in vertebrates. NE and EP signaling was fully established in vertebrate species, a response that provided additional support for the motor activation process and was better suited to a more sophisticated mobile lifestyle. Thus, this chemical communication system was also expanded and developed to promote complex motivational processes, including reward, pleasure, and pain. This hypothesis provides a cogent explanation for the well-established links between motor activities and emotional processes.

This leads to our final hypothesis, namely, that catecholamine-based processes, including emotion and cognition, were developed as a coping strategy and another means to focus on the activation of motor processes, making them more complex and thus, highly valued as a survival strategy. Collectively, these evolutionary developments promote the prolonged survival of numerous vertebrate species [2,43,44]. However, since these processes link mobility to cognition via the same chemical pathways, alterations in these pathways can have a strong impact and make them vulnerable, e.g., as potential strategic viral targets.

## Figures and Tables

**Figure 1 biology-12-00080-f001:**
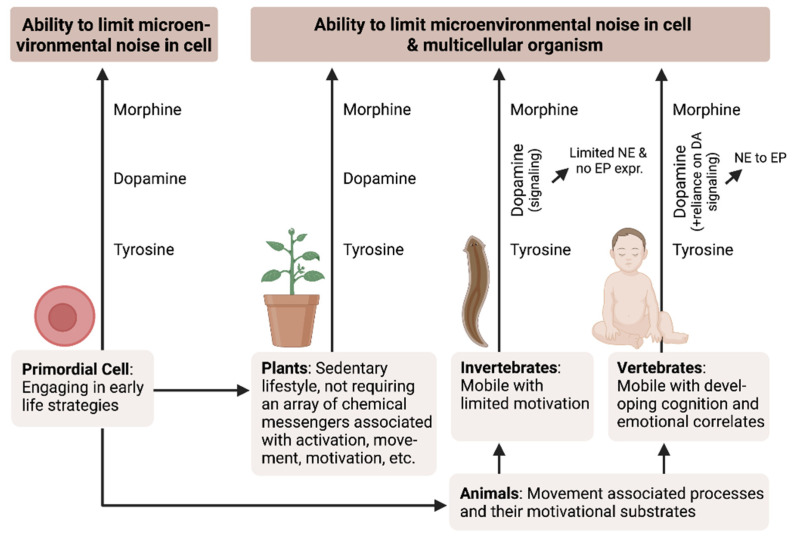
Changes observed in response to evolutionary constraints. While dopamine (DA) may be detected in plant and primordial cells, these systems do not rely on catecholamine signaling pathways. By contrast, DA is a prominent signaling agent in invertebrate animal species. In certain comparatively long-lived invertebrates, DA is modified to generate dopamine ß-hydroxylase. In vertebrates, the morphine biosynthetic enzyme, catechol O-methyl transferase (COMT), has been “retrofitted” and facilitates the conversion of norepinephrine (NE) to epinephrine (EP). This relationship is further highlighted by observations documenting the colocalization of catecholamine with endogenous morphine in the brain. Of note, DA has been identified as an endogenous morphine precursor; likewise, morphine-induced constitutive NO serves to downregulate excitatory processes and scavenge free radicals that result at least in part as a byproduct of catecholamine signaling (references: [1,2,10,11,12,13,14,15,16,17,18,19,20,21]).

**Figure 2 biology-12-00080-f002:**
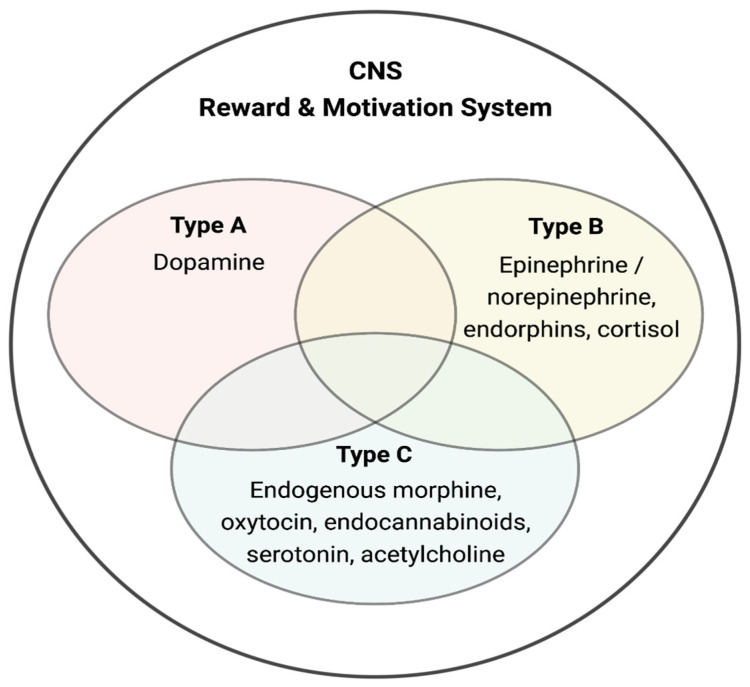
Type A–C motivation systems. These subsystems (A → appetitive motivation; B → stress/threat avoidance; C → relaxation/quiescence) share a common process (enzymatic-neurobiological cascade)—mechanisms that converge on the CNS motivational and reward system that includes critical neurotransmitters and distinctive regions (i.e., specific reward messengers, among others; references: see text and [44]).

**Figure 3 biology-12-00080-f003:**
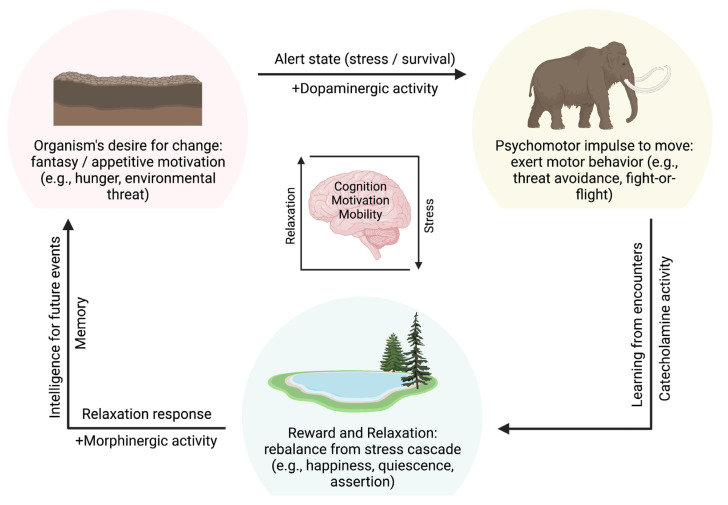
Functional significance of the three motivation systems from an evolutionary biological perspective. Appetite/approach, avoidance/aversion, and assertion/quiescence follow in a motivational order that connects cognition with mobility: Mobile animals evolved an “appetite” for change, followed by a psychomotor impulse to move and behaviorally implement the actual change, which also required the ability to successfully overcome challenges along the way. The ability to recover from this stress was critical, also learning from its “adventures” for future encounters, eventually terminating this neurobiological sequence (motivational “circle”) by returning to normal and being “happy” (references: see text).

## Data Availability

Not applicable.

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
