# Peer review of "Mobility Coupled with Motivation Promotes Survival: The Evolution of Cognition as an Adaptive Strategy"

_biology, 2023, doi:10.3390/biology12010080_

Round 1

Reviewer 1 Report

Some background information regarding morphine and dopamine signalling in plants would further enhance the article.

Author Response

Thank you for your helpful comment ("Some background information regarding morphine and dopamine signalling in plants would further enhance the article"). WE HAVE ADDED A NEW PARAGRAPH ON MORPHINE/DA SIGNALING IN PLANTS.

Reviewer 2 Report

The review highlights the biosynthetic interweaving between tyrosine, dopamine, endogenous morphine, adrenaline and noradrenaline which unravels through evolution starting from the plant world up to invertebrates and vertebrates. This had already been highlighted in a previous work by the authors: INTERNATIONAL JOURNAL OF MOLECULAR MEDICINE 20: 837-841, 2007

In this new review, based on previous literature also by the same authors, the evolutionary framework of the neurotransmitters mentioned above and chemically is correlated with the appearance of increasingly evolved abilities related to mobility up to the development of cognitive abilities in vertebrates.

The review is well constructed and brings numerous data from the literature and authors to support the hypotheses supported.

A few more schemes would help the review: p6 line 233-243

Minor points: some bibliographic entries with more than three authors are reported with the wording after the three authors et al. Other bibliographic entries carry numerous authors. References should be standardized.

Fig 3 cited in the text does not appear in the text: pag 3 line 121 and 124

The bibliographic entry 165 does not exist : page 5

Author Response

Thank you for your helpful comments:

1) "A few more schemes would help the review: p6 line 233-243." WE ADDED A NEW FIGURE 3

2) "Minor points: some bibliographic entries with more than three authors are reported with the wording after the three authors et al. Other bibliographic entries carry numerous authors. References should be standardized." DONE - ALL REFERENCES ARE STANDARDIZED NOW

3) "Fig 3 cited in the text does not appear in the text: pag 3 line 121 and 124." WE DELETED THAT CITATION & CHECKED ALL OTHER CITATIONS FOR FIGURES THROUGHOUT THE TEXT; WE ALSO ADDED A NEW FIGURE 3, WHICH IS NOW CORRECTLY REFERENCED IN THE MAIN TEXT

4) "The bibliographic entry 165 does not exist : page 5." CORRECTED